# What Differentiates Poor and Good Outcome Psychotherapy? A Statistical-Mechanics-Inspired Approach to Psychotherapy Research

**Giulio de Felice** [1,2], **Franco F. Orsucci** [2,3], **Andrea Scozzari** [4], **Omar Gelo** [5,6],
**Gabriele Serafini** [4,7], **Silvia Andreassi** [1], **Nicoletta Vegni** [8], **Giulia Paoloni** [1], **Gloria Lagetto** [5],
**Erhard Mergenthaler** [9] and **Alessandro Giuliani** [10,*]

[1]   Department of Dynamic and Clinical Psychology, Sapienza University of Rome, 00185 Roma, Italy;
      giulio.defelice@uniroma1.it (G.d.F.); silvia.andreassi@uniroma1.it (S.A.); g.paoloni@uniroma1.it (G.P.)
[2]   Department of Psychology, Niccolò Cusano Italian University in London, London E14 9TS, UK;
      franco.orsucci@nciul.ac.uk
[3]   Department of Psychoanalysis, University College of London, London WC1E 6BT, UK
[4]   Faculty of Economics, Niccolò Cusano University of Rome, 00166 Roma, Italy;
      andrea.scozzari@unicusano.it (A.S.); gabriele.serafini@unicusano.it (G.S.)
[5]   Department of History, Society and Human Studies, University of Salento, 73100 Lecce LE, Italy;
      omar.gelo@unisalento.it (O.G.); gloria.lagetto@studenti.unisalento.it (G.L.)
[6]   Faculty of Psychotherapy Science, Sigmund Freud University, A-1020, Vienna 1050, Austria
[7]   Department of Economics, NC Italian University in London, London E14 9TS, UK
[8]   Faculty of Psychology, Niccolò Cusano University of Rome, 00166 Roma, Italy; nicoletta.vegni@unicusano.it
[9]   Clinic of Psychosomatic Medicine and Psychotherapy, University of Ulm, 89081 Ulm, Germany;
      erhard.mergenthaler@uni-ulm.de
[10]  Department of Environment and Health, Istituto Superiore di Sanità (Italian National Institute of Health)
      Roma, Viale Regina Elena 299, 00161 Roma, Italy
*    Correspondence: alessandro.giuliani@iss.it

**Abstract:** Statistical mechanics investigates how emergent properties of macroscopic systems (such as temperature and pressure) relate to microscopic state fluctuations. The underlying idea is that global statistical descriptors of order and variability can monitor the relevant dynamics of the whole system at hand. Here we test the possibility of extending such an approach to psychotherapy research investigating the possibility of predicting the outcome of psychotherapy on the sole basis of coarse-grained empirical macro-parameters. Four good-outcome and four poor-outcome brief psychotherapies were recorded, and their transcripts coded in terms of standard psychological categories (abstract, positive emotional and negative emotional language pertaining to patient and therapist). Each patient-therapist interaction is considered as a discrete multivariate time series made of subsequent word-blocks of 150-word length, defined in terms of the above categories. "Static analyses" (Principal Component Analysis) highlighted a substantial difference between good-outcome and poor-outcome cases in terms of mutual correlations among those descriptors. In the former, the patient's use of abstract language correlated with therapist's emotional negative language, while in the latter it co-varied with therapist's emotional positive language, thus showing the different judgment of the therapists regarding the same variable (abstract language) in poor and good outcome cases. On the other hand, the "dynamic analyses", based on five coarse-grained descriptors related to variability, the degree of order and complexity of the series, demonstrated a relevant case-specific effect, pointing to the possibility of deriving a consistent picture of any single psychotherapeutic process. Overall, the results showed that the systemic approach to psychotherapy (an old tenet of psychology) is mature enough to shift from a metaphorical to a fully quantitative status.

**Keywords:** psychotherapy; complex systems; statistical mechanics; process of change; nonlinear dynamics

## 1. Introduction

In the literature, there have been many examples aimed at finding coarse-grained descriptors able to explain the behavior of complex systems composed of several different elements. Statistical thermodynamics has emphasized the importance of focusing on the dynamics of the degree of order of a system [1]. This approach can be extended to any scientific field, posited that we get a sensible measure of system autocorrelation [2–5].

In econometrics, Gorban and colleagues [2] showed that a market's change occurs after a critical period (tipping point) in which both the internal correlation and variability of the system reach a peak value. In biology, many studies [6–8] showed the usefulness of looking at biological systems from the perspective of statistical mechanics, that is, focusing on the mutual correlations among system descriptors. This scientific stance takes the name of "middle-out" approach since it focuses on a mesoscopic level maximizing the correlations among system descriptors. In other words, this approach lies "in the middle" between pure "bottom-up" (the causally relevant layer is the microscopic one) and "top-down" (the causally relevant layer is where general laws are defined) approaches [9,10].

Along these lines of thought, in the psychotherapy research literature, Schiepek and colleagues [11] formulated an empirical dynamic descriptor that predicts the therapeutic change. It is obtained by the multiplication of the distribution and fluctuation of a given signal (for a detailed description, see reference [11]), and can be intended as a measure of system variability (for a review, see reference [12]). Analogously to what expected [2–5], a peak of "dynamic complexity" was usually found to precede a therapeutic change or restructuring. Clinically, this behavior corresponds to the observation of something new in the patient's in-session narratives or in some of his/her behavioral traits outside the clinical room before the occurrence of an important insight [13,14]. Considering the contributions based on the application of the dynamic systems approach to psychotherapy research (for a review on the application of dynamical systems theory or complexity science to psychotherapy research see reference [12]), some attempts have been made to empirically identify desirable attractors in the psychotherapeutic process. Tschacher and Ramseyer [15] approached the problem in terms of stable dynamic patterns of different variables pertaining to the psychotherapeutic process. Other authors tried to detect pattern formation following the rationale of self-organization (e.g., reference [16]); Gumz and colleagues [17] investigated critical fluctuations characterizing moments of instability in the psychotherapeutic system. In these works, there are some attempts, albeit scattered, to investigate the evolution of psychotherapy through coarse-grained empirical indices of entropy, autocorrelation and variability. Despite that, empirical proof of the possibility to predict the evolution of psychotherapy by means of trans-theoretical indices of order, variability, and complexity (the three most widespread macro-parameters employed to investigate a system's behavior) has never been obtained.

Hence, the present work is grounded on the following two research questions: (1) What are the relevant macro-parameters to describe psychotherapy and how do they interact? (2) Are these macro-parameters able to differentiate good and poor outcome psychotherapy?

The present work represents the first empirical effort to provide an answer to these questions by applying a statistical-mechanics-inspired approach to psychotherapy.

## 2. Material and Methods

### 2.1. Sample

The sample analyzed was drawn from the York Depression Study I, a randomized clinical trial to assess the efficacy of brief experiential therapy (client-centered therapy (CCT) and emotion-focused therapy (EFT)) for depression [14]). The original sample was comprised of 17 CCT and 17 EFT. Hence,

for the present study, we focused on these cases including their transcripts. Initially, we selected the six best-outcome cases (CCT = 3; EFT = 3) and the six worst-outcome cases (CCT = 3; EFT = 3) based on the Reliable Change Index (i.e., RCI; [18]) of the Beck Depression Inventory (BDI; [19]). Further, four cases (3 = CCT and 1 = EFT) were excluded due to some missing sessions. In so doing, our final sample was comprised of four good-outcome (1 = CCT and 3 = EFT) and four poor-outcome (2 = CCT and 2 = EFT) cases, for a total of N = 8 treatments. For the good outcome cases, the BDI pre-post change was from 30 to 5, from 25 to 3, from 35 to 4, and from 21 to 12. For the poor outcome cases, it was from 23 to 22, from 15 to 13, from 19 to 19, and from 13 to 9 (for more details, see also [16]).

Patients: Patients were one man and seven women with a mean age of 37.08 years who met the criteria for major depressive disorder (MDD) on the Structured Clinical Interview for DSM-III-R (SCID; [17]).

Therapists: Therapists were seven women and one man with an average of approximatively 5.5 years of therapeutic experience and 24 weeks of training in experiential psychotherapy [20]. All therapists were monitored for adherence through video recordings and weekly supervisions.

Treatments: CCT, emphasizes empathy, positive regards, and congruence [21,22], and EFT integrates CCT with "process-directive gestalt and experiential interventions" for the resolution of dysfunctional cognitive-affective processing [22]. The treatment length was between 15 and 20 sessions (mean = 17.62, st.dev. = 1.38) for a total of 141 sessions.

## 2.2. Measures

The semantic production of the eight brief psychotherapies was coded according to Mergenthaler's Therapeutic Cycles Model (TCM; [23–25], a computer-assisted deductive content analytic tool which breaks the transcript down into chunks of 150 word-blocks and then analyses them according to three different dictionaries: (a) Positive emotional tone (POS), (b) negative emotional tone (NEG), and (c) abstraction (AB). The first two contain adjectives, verbs, or adverbs with a positive or negative valence (e.g., happy, sad; agree, disagree; hug, abandon; incredible, astonished). The third contains abstract words (e.g., year, hour, accident, soul, wedding). All sessions were transcribed according to the international standards [25]. The TCM automatically assesses the relative frequency of the three dictionaries per each word-block.

In short, the dataset has 6 variables—the relative frequencies of the three vocabularies pertaining to patient and therapist of each therapy (i.e., 8 different therapeutic couples, no therapist carried out more than one therapy)—and as statistical units the word-blocks in temporal order.

## 2.3. Data Analysis

We investigated the correlation structure of the data set using Principal Component Analysis (PCA, [26,27]). The strategy of analysis stems from the hypothesis that good and poor outcome cases corresponded to two different correlation structures imposed on the 6 variables. This is consistent with the fact that the same variable takes a different meaning (and consequently a different pattern of correlation) in good and poor outcome contexts. Thus, given the extracted components are eigenvectors of the correlation matrix [26], we expect the distinct PCA solutions pertaining to poor-outcome cases, good-outcome cases and good-and-poor-outcome cases taken together not to be super-imposable, for the changes in the correlation structure linked to different outcomes [27]. Moreover, given that the principal components of a specific dataset are each other orthogonal by construction, the discovery of mutual correlations between components of those three solutions allows us to highlight "hidden variables" which have the same latent meaning even if endowed with a different loading structure. The actual steps of the analysis were scheduled as follows, following the scheme set forth in reference [27]:

(1)　Three independent PCAs are performed: (a) PCA of poor-outcome cases (6 variables); (b) PCA of good-outcome cases (6 variables); and (c) PCA of good-and-poor-outcome cases taken together with 12 (6 + 6) variables.

(2)　The component scores of the 24-variable case (i.e., 12 + 6 + 6) are scrutinized by means of mutual Pearson correlations. This procedure allows us to gather two pieces of crucial information. On the

one hand, the component scores pertaining to (a) or (b) (subset) cases that scale with the same component scores of (c) (whole set) point to latent factors common to good and poor outcome cases (below called "mixed"). On the other hand, the component scores pertaining to (c) that scale only with one of the (a) and (b) subsets are peculiar to either poor or good outcome cases (below called "pure").

These results can be helpful in understanding the differences and analogies pertaining to good and poor outcome datasets. However, as they depict the latent dimensions of a given data matrix, with no explicit reference to time course, they produce static results.

In order to understand the dynamics pertaining to good and poor outcome cases as well as their possible differences and analogies, we need to go back and study the original data, recovering their temporal dimension, changing our level of analysis from the aggregation of cases to single patient-therapist dynamics. Building upon the widely recognized link between the onset of transitions and the increase in the autocorrelation of the system at hand [2], we computed different statistical indices of temporal correlation on the psychotherapy time series.

We divided the original data matrix of each patient (i.e., 6 linguistic variables as columns and consecutive word-blocks as rows) into sub-matrices: One sub-matrix for each psychotherapeutic session. The sub-matrices were the input for the computation of five different statistical indices, widely used to forecast transitions in temporal series:

(a) Canonical Correlation Coefficient; (b) Percentage of variance explained by the first principal component; (c) Sum of Pearson correlation coefficients higher than |0.25|; (d) Standard deviation of Pearson coefficients, and (e) Shannon entropy on eigenvalues [3].

The usefulness of these indices in anticipating critical transitions was assessed in many different fields spanning from economics to biology and chemistry [2], consistently with the universal character of statistical mechanics. The rationale behind these measures (see Appendix A) is the existence of common dynamics relating to both the correlation and variability of a system approaching a critical transition [4].

The "canonical correlation coefficient" corresponds to the highest canonical correlation between patient (X variable subset) and therapist (Y variable subset) descriptors and, thus, monitors the strength of the interaction between therapist and patient along the process.

The "percentage of variance explained by the first principal component" is a measure of the degree of correlation for all the variables (both therapist- and patient-related). A very similar interpretation holds for the "sum of Pearson correlation coefficients higher than |0.25|", but in this case, the estimation of the amount of correlation is limited to middle-to-high correlations.

The "standard deviation of Pearson correlation coefficients" points to another feature of correlation dynamics: its variation in time, and, thus, it is supposed to be more sensitive to the presence of "transitions" along the process. The last index, "Shannon entropy of eigenvalues" focuses on the dimensionality of the time series on the phase space; a high entropy points to the lack of any dominant correlation flux.

A routine in Matlab was developed to compute such variability and correlation indices on every single case with a time window corresponding to one session. Then, the extracted measures were examined by a Principal Component Analysis to understand their latent dimensions. Eventually, their component scores were included as dependent variables of three General Linear Models to check the existence of peculiar case-specific correlation dynamics (section "Dynamic Analyses"). The discovery of such relations would be proof that the above formalization grasps the uniqueness of each psychotherapeutic process, thus constituting its sensible quantitative description.

## 3. Results

### 3.1. Static Analyses

From the correlation matrix of the original dataset made of both good- and poor-outcome cases, it was not possible to obtain any significant information. The pairwise Pearson coefficients, for the

variables AB (abstract language), POS (positive language), and NEG (negative language) of therapists and patients, were near to zero, that is, apparently linear independent.

The PCA confirmed this observation displaying a flat eigenvalues distribution (Table 1), so pointing to the lack of a relevant shared correlation structure at this level of analysis. This implies that the adopted coding system is made of largely independent categories; in other words, AB, POS, and NEG as such are well-defined, not overlapping concepts.

**Table 1.** Principal Component Analysis (PCA) of good and poor outcome cases. Percentages of variance explained by each component for good-outcome and poor-outcome cases are shown.

| Component | PCA Good-Outcome Cases. Eigenvalues, % of Variance Explained. | PCA Poor-Outcome Cases. Eigenvalues, % of Variance Explained. |
|---|---|---|
| 1 | 23.192 | 20.742 |
| 2 | 18.112 | 18.303 |
| 3 | 17.114 | 16.896 |
| 4 | 15.707 | 15.752 |
| 5 | 14.393 | 15.360 |
| 6 | 11.482 | 12.947 |

In order to grasp the potential differences in the correlation structures of good- and poor-outcome cases, we deepened their study by performing three different principal component analyses at different definition levels: (a) PCA of poor-outcome cases, (b) PCA of good-outcome cases, and (c) PCA of poor-and-good-outcome cases taken together. Then, the component scores obtained were scrutinized by a Pearson correlation matrix to identify (see Methods and Data Analysis) "mixed" and "pure" components across good and poor-outcome cases. The latter components, peculiar to either good or poor-outcome cases, will show the idiosyncrasies of patient-therapist interactions in the two outcome classes. Components 1, 5, and 6 of good-outcome and poor-outcome cases pertain to this latter category, while components 2, 3, and 4 to the former (Table 2).

Good and poor principal component solutions are substantially different: when we merge the two groups of descriptors by performing a global PCA with 12-variables, some of their peculiarities (i.e., "pure" good- or poor-outcome components) do emerge (in bold and on a red background, Table 2). Metaphorically, if our outcome groups had been fluids, we would have been able to observe the behavior of something like oil and water, or a very poor vodka Martini: even if we keep shaking it, there would always be some unstirred components (components 1, 5 and 6) (see [27]).

Now, in order to maximize the differences between good-outcome and poor-outcome cases, we performed the Pearson correlation matrices partializing the effect of the "mixed" components, 2, 3 and 4. Table 3 shows the main differences between poor and good outcome cases.

The differences (in bold on a red background) in the correlation matrices between good-outcome and poor-outcome cases are now evident. The most obvious one concerns the dynamic in which the patient makes use of abstract language (AB), interpreted by the therapist very positively in poor-outcome cases and very negatively in good-outcome cases (see the opposite sign of correlation of AB with POS and NEG therapist variables in the two subsets). In the latter, the use of an abstract language is probably considered as a patient's defense mechanism to be addressed, a way to avoid speaking of his/her own emotions. On the contrary, in poor-outcome cases, it is interpreted as a positive sign of a supposed critical thinking and reflection on his/her pathological condition.

This interpretation is consistent with the use of positive and negative emotional languages that are inversely proportional to abstraction only in poor-outcome patients.

The defense mechanism of rationalization is a way to protect the mind from painful feelings using an abstract, intellectual, and often concrete attitude in dealing with them. The good-outcome therapists seem to respond promptly to that by addressing the emotional content lying under the surface of the psychotherapeutic field (i.e., use of the therapist's negative emotional language). In the poor-outcome cases the therapists seem less able to address the dynamic of rationalization, or, on the other hand, the poor-outcome patients make an extreme use of it impeding their own clinical progress.

**Table 2.** Pearson Correlation Matrix between the three PCAs' component scores (here, for the sake of simplicity, only the area with some significant correlations is shown). The components that scale with only one component of PCA (c) (e.g., only good-comp1 scales with component (1), are given in bold on a red background. They represent peculiar features of either good or poor-outcome cases. The other components are coupled (e.g., both good-comp2 and poor-comp2 scale with the same component) and scale with more than one component of PCA (c). They don't represent specific features of either good or poor outcome cases.

| Components PCA (c) | Good-comp1 | Good-comp2 | Good-comp3 | Good-comp4 | Good-comp5 | Good-comp6 | Poor-comp1 | Poor-comp2 | Poor-comp3 | Poor-comp4 | Poor-comp5 | Poor-comp6 |
|---|---|---|---|---|---|---|---|---|---|---|---|---|
| 1 | **0.991 ** | 0.008 | 0.003 | 0.004 | 0.012 | 0.004 | 0.088 ** | 0.127 ** | −0.015 | 0.025 | −0.010 | 0.081 ** |
| 2 | 0.075 ** | −0.029 | −0.010 | 0.045 ** | −0.008 | −0.051 ** | **0.995 ** | 0.020 | −0.006 | 0.014 | −0.006 | 0.015 |
| 3 | −0.070 ** | **0.710 ** | 0.015 | 0.055 ** | 0.058 ** | −0.011 | 0.013 | **0.694 ** | 0.143 ** | −0.137 ** | 0.063 ** | −0.024 |
| 4 | −0.076 ** | **−0.568 ** | −0.194 ** | 0.201 ** | 0.067 ** | −0.012 | −0.033 * | **0.658 ** | −0.341 ** | 0.200 ** | −0.051 ** | 0.023 |
| 5 | −0.015 | −0.254 ** | **0.800 ** | 0.062 ** | −0.047 ** | −0.030 | −0.005 | 0.177 ** | **0.495 ** | 0.139 ** | −0.085 ** | 0.049 ** |
| 6 | −0.001 | 0.170 ** | **0.560 ** | 0.047 ** | 0.077 ** | 0.023 | 0.007 | −0.064 ** | **−0.779 ** | −0.103 ** | 0.027 | 0.020 |
| 7 | −0.018 | 0.241 ** | −0.044 ** | **0.708 ** | −0.134 ** | −0.047 ** | −0.025 | −0.163 ** | −0.011 | **0.630 ** | −0.047 ** | 0.131 ** |
| 8 | 0.010 | −0.058 ** | 0.030 | −0.001 | 0.302 ** | 0.079 ** | 0.006 | −0.017 | 0.051 ** | 0.158 ** | **0.940 ** | −0.025 |
| 9 | −0.001 | 0.119 ** | 0.026 | **−0.642 ** | −0.154 ** | −0.021 | 0.017 | 0.063 ** | −0.102 ** | **0.683 ** | −0.058 ** | −0.174 ** |
| 10 | 0.015 | 0.037 * | −0.021 | −0.002 | **0.914 ** | −0.030 | 0.005 | −0.068 ** | 0.058 ** | 0.144 ** | −0.307 ** | −0.130 ** |
| 11 | −0.042 * | 0.024 | −0.027 | −0.174 ** | 0.103 ** | −0.274 ** | −0.006 | 0.005 | −0.008 | 0.037 * | 0.004 | **0.933 ** |
| 12 | −0.005 | 0.002 | −0.010 | −0.047 ** | 0.030 | **0.953 ** | 0.027 | 0.011 | 0.016 | 0.042 * | −0.069 ** | 0.238 ** |

**. Correlation is significant at the 0.01 level (2-tailed). *. Correlation is significant at the 0.05 level (2-tailed).

**Table 3.** Partial Correlations of good (top) and poor (bottom) outcome cases. Their main differences are given in bold and on a red background.

| Control Variables (Good-comp2, 3 and 4) | AB Therapist (Good Cases) | POS Therapist (Good Cases) | NEG Therapist (Good Cases) | AB Patient (Good Cases) | POS Patient (Good Cases) | NEG Patient (Good Cases) |
|---|---|---|---|---|---|---|
| AB therapist | 1.000 | −0.259 ** | −0.460 ** | −0.463 ** | −0.114 ** | 0.780 ** |
| POS therapist | | 1.000 | −0.541 ** | **−0.714 **** | **−0.183 **** | **−0.583 **** |
| NEG therapist | | | 1.000 | **0.918 **** | −0.346 ** | 0.186 ** |
| AB patient | | | | 1.000 | **0.046 **** | **0.078 **** |
| POS patient | | | | | 1.000 | −0.473 ** |
| NEG patient | | | | | | 1.000 |
| **Control Variables (Poor-comp2, 3 and 4)** | **AB therapist (poor cases)** | **POS therapist (poor cases)** | **NEG therapist (poor cases)** | **AB patient (poor cases)** | **POS patient (poor cases)** | **NEG patient (poor cases)** |
| AB therapist | 1.000 | −0.170 ** | −0.064 ** | −0.637 ** | −0.323 ** | 0.977 ** |
| POS therapist | | 1.000 | −0.424 ** | **0.831 **** | **−0.559 **** | **−0.296 **** |
| NEG therapist | | | 1.000 | **−0.072 **** | −0.343 ** | 0.148 ** |
| AB patient | | | | 1.000 | **−0.418 **** | **−0.680 **** |
| POS patient | | | | | 1.000 | −0.354 ** |
| NEG patient | | | | | | 1.000 |

**. Correlation is significant at the 0.01 level (2-tailed). *. Correlation is significant at the 0.05 level (2-tailed).

## 3.2. Dynamic Analyses

We divided the original data matrix of each patient (i.e., 6 linguistic variables as columns and consecutive word-blocks as rows, around 800 word-blocks per patient) into sub-matrices, one sub-matrix for each psychotherapeutic session. Then we computed, from the above sub-matrices, five different indices used as possible transition signatures: (a) Canonical correlation coefficients; (b) Percentage of explained variance by the first component; (c) Sum of the Pearson correlation coefficients higher than |0.25| (called "Gorban's G" [2]); (d) Standard deviation of Pearson coefficients, and (e) Shannon Entropy on Eigenvalues. A routine in Matlab was developed to compute the aforementioned statistical indices on every single case with a time window corresponding to one session. Then, these results were examined with a PCA in order to investigate their latent dimensions (Table 4).

**Table 4.** Principal Component Analysis of the five dynamic indices after their application on the eight single cases. Three components (in bold) were retained by the scree test criterion. The most important descriptors of each component are given in bold (see "Components' Loadings").

| Component | Eigenvalue | Difference | Proportion | Cumulative |
|:---:|:---:|:---:|:---:|:---:|
| 1 | **2.36** | **0.73** | **0.47** | **0.47** |
| 2 | **1.62** | **1.11** | **0.32** | **0.79** |
| 3 | **0.50** | **0.21** | **0.10** | **0.89** |
| 4 | 0.29 | 0.08 | 0.05 | 0.95 |
| 5 | 0.21 | | 0.04 | 1.00 |

| Components' Loadings | | | |
|:---:|:---:|:---:|:---:|
| | **Component 1** | **Component 2** | **Component 3** |
| Canonical Correlation | **0.77** | 0.46 | −0.31 |
| Shannon Entropy | −0.30 | **0.86** | 0.11 |
| Gorban's G | **0.86** | 0.29 | −0.18 |
| Standard Deviation | **0.79** | 0.05 | **0.60** |
| Variance I Component | 0.52 | **−0.75** | −0.08 |

Component 1 appears to be strictly linked with the Pearson correlation standard deviation, the average amount of correlation, and canonical correlation of a given matrix. This component allows for a clinical interpretation, being a proxy of those periods of time in which a patient does something new prior to a proper insight (i.e., often called "second order change"). It can sometimes happen that the patient, especially at the beginning of psychotherapy, opens a "forgotten" drawer to see old pictures of herself/himself or her/his family, regaining pleasure in old habits like doing sport or meeting old friends, or travels to places associated with her/his past. In these conditions, her/his behavioral variability increases as well as the correlation robustness of his/her personal history: S/he becomes able to narrativize it, and, thus, her/his personal identity acquires robustness and coherence. It is a dimension that can be summarized with the following polarities: order-variability.

Component 2 appears to be linked with the Shannon Entropy on the eigenvalues and the variance explained by the first component. When the former has a peak, we have a very "flat" scree plot with all the principal components (which can be matched to the notion of "order parameters") pulling the data matrix into their respective directions, with almost equal strength. Alternatively, when the variance explained by the first component has a peak, the slope of the scree plot increases a great deal, giving rise to a clear organization in the system. Clinically, this latest picture has often been called "first order change", in which a patient is now able to describe the reasons, for example, of a relational attitude s/he was aware of but could not explain. In short, the patient's personality does not re-organize itself as in the case of second order changes, but it attains more coherence. That is why we do not observe an increase in its standard deviation here (i.e., the necessary precondition of a personality re-organization). It is a dimension that can be summarized with the following polarities: elementary-complex.

Component 3 is linked with the standard deviation of Pearson coefficients independently from the system's degree of order.

Summing up:

- Component 1: The higher the component score, the higher the relational consistency between therapist and patient (canonical correlation), its correlation robustness and variability. A dimension described with the polarities of order-variability.
- Component 2: The higher the component score, the higher the complexity of the system (the more negative the correlation with the amount of variance explained by the first component, the flatter the spectrum of eigenvalues). A dimension described with the polarities of elementary-complex.
- Component 3: The higher the component score, the higher the emergence of "low" and "high" correlation phases along the process.

Then, the component's scores were included as dependent variables of three General Linear Models with the different therapeutic dyads as a source of variation. This allows to investigate if the three components describe the peculiarities of every single psychotherapeutic process, or, in the case of lack of significance, if the 8 therapeutic series are not discriminable, and consequently, the coarse-grained descriptors chosen are not their sensible quantification. In so doing, we tested the hypothesis to find some macro-parameters that significantly explain the temporal peculiarities of each psychotherapeutic process independently of their different theoretical orientations (Table 5).

**Table 5.** The GLM procedure. The variables which are statistically significant in describing each psychotherapeutic process are given in bold.

| The GLM Procedure | | | | | |
|---|---|---|---|---|---|
| **Dependent Variable: Component 1** | | | | | |
| **Source** | **DF** | **Sum of Squares** | **Mean Square** | **F Value** | **Pr > F** |
| Model | 7 | 35.47 | 5.06 | **6.45** | **<0.0001** |
| Error | 133 | 104.52 | 0.78 | | |
| Corrected Total | 140 | 140 | | | |
| | | **R-Square** | **Coeff. Var.** | **Root MSE** | **Mean Component 1** |
| | | 0.25 | $-5.4 \times 10^{14}$ | 0.88 | 0 |
| **Dependent Variable: Component 2** | | | | | |
| **Source** | **DF** | **Sum of Squares** | **Mean Square** | **F Value** | **Pr > F** |
| Model | 7 | 17.79 | 2.54 | **2.77** | **0.010** |
| Error | 133 | 122.21 | 0.91 | | |
| Corrected Total | 140 | 140 | | | |
| | | **R-Square** | **Coeff. Var.** | **Root MSE** | **Mean Component 2** |
| | | 0.12 | $-4.47 \times 10^{15}$ | 0.95 | 0 |
| **Dependent Variable: Component 3** | | | | | |
| **Source** | **DF** | **Sum of Squares** | **Mean Square** | **F Value** | **Pr > F** |
| Model | 7 | 8.47 | 1.21 | 1.22 | 0.29 |
| Error | 133 | 131.53 | 0.99 | | |
| Corrected Total | 140 | 140 | | | |
| | | **R-Square** | **Coeff. Var.** | **Root MSE** | **Mean Component 3** |
| | | 0.06 | $-2.85 \times 10^{15}$ | 0.99 | 0 |

The hypothesis is verified for components 1 and 2, which can be viewed as statistically significant descriptors of the psychotherapeutic processes. Their variability is case-specific (that is, it is dependent

on being part of a given therapeutic dyad), and they represent statistically significant macro-parameters of the psychotherapeutic process, independently from the therapeutic approach.

Concerning significant differences in those indices between poor-outcome and good-outcome cases, the former showed greater linguistic redundancy (more variance explained by the first principal component, $p = 0.003$) and less variability (less standard deviation of Pearson correlation coefficients, $p = 0.011$). Regarding the differences between psychotherapists and patients, the former showed greater linguistic variability (higher standard deviation of Pearson correlation coefficients, $p < 0.0001$), and less redundancy (less variance explained by the first principal component, $p = 0.001$).

Going back to the original descriptors, the only two variables showing a statistically significant difference between good- and poor- outcome cases were the standard deviation of Pearson correlation coefficients (Std, Component1) and the variance explained by the first principal component (Variance I Component, Component 2). Over the 150 blocks for each group, the mean and standard deviation of Std were 0.525 (0.05) and 0.500 (0.05) for good and poor groups respectively; while for the Variance I Component, the statistics were 49.62 (7.50) and 52.4 (8.9) for good and poor groups, respectively.

Despite reaching statistical significance, this is in any case a weak difference, but, given the exploratory nature of this work and the poor numerosity of the sample size, it can be considered as a promising initial result.

## 4. Conclusions

By means of "static analyses" we were able to highlight significant differences between good- and poor-outcome cases concerning their latent correlation structures. These results pointed to a shift of meaning of the adopted psychological dictionaries dependent on the correlation structures of the two outcome classes. The most evident difference was linked to the patient's use of abstract language, interpreted very positively in poor-outcome cases and very negatively in good-outcome cases. This observation is closely associated to the use of positive and negative emotional languages inversely proportional to abstraction only in poor-outcome patients. This allows us to affirm that, despite the limit of the small sample size, we can give an affirmative answer to both the research questions highlighted in the introduction. These are in any case still partial, asking for a more thorough dynamical approach which we will investigate in the following studies. An open question is whether the poor-outcome patients are more inclined to rationalization, or, alternatively, the poor-outcome therapists are less able to address this defense mechanism. In the first case, we can conceive a more prolonged therapy or an exclusion of such patients from the brief therapy protocols; in the second case, it should be highlighted the importance of clinical, or clinical and empirical (feedback derived from empirical data), supervision.

However, it is worth stressing that the comparison between the three PCA solutions clearly showed that the same variable changed its meaning according to its correlation structure with the other variables. This is a classical feature of any proper "system", from chemistry (a hydrogen atom bonded to oxygen in water has different properties when bonded to carbon in a methane molecule), to ecology (the same species that contributes positively to the system equilibrium can be a threat to the ecological balance when they are placed in a different environment). Thus, the results emphasize the crucial importance of grounding our clinical and research investigations on such a systemic view also in the domain of psychotherapy [28–31].

Regarding the "dynamic analyses", it is worth underlining that, while they are much more convincing to outline a usable picture of the ongoing psychotherapy process than the "static analyses", such a dynamical approach is much more difficult to accomplish. The difficulty derives from many sources such as the contingent quality of the individual episodes along the process, the coarse-grained quality of the three dictionaries, and clearly, the impossibility of taking into consideration potentially crucial "context variables" like the relative empathy established in patient-therapist dyads, the socio-demographic status of the patient, and so forth. The observed statistical significance, given the

relevance of the effect, can only be interpreted as an indication of the interest in pursuing such kinds of analysis beyond this preliminary explorative phase.

Despite these limitations, the results show the possibility to describe the psychotherapy process, independently from the theoretical approach, with two quantitative macro-parameters, namely, order-variability (PC1) and elementary-complex (PC2). It is worth noting that these two macro-parameters give a quantitative value to concepts often present, even if implicitly, in the therapist's mind. Are the narratives of a given patient rigid and fixed or are they flexible and adaptable? Is there some new element in them or are they always going around the same anxiety (stationary attractor)? Is the narrative rich or impoverished? Is the patient's thought symbolic or concrete? These are all typical clinical questions implicitly concerning the system's variability, the degree of order and complexity or richness of information, and this work represents, to our knowledge, the first successful effort to translate them empirically in the domain of psychotherapy. This work is part of a larger project including quantitative and qualitative analyses of clinical transcripts in order to understand change and its trajectories in clinical and empirical terms. Here we showed the results stemmed from the quantitative analyses over the psychotherapeutic transcripts. In the following study, we plan to apply a multi-layer explorative approach by means of complex network analyses. The systemic approach, often widely claimed in psychotherapy, promises to become operational.

**Author Contributions:** Conceptualization, G.d.F. and A.G.; Methodology, G.d.F., A.G., A.S.; Formal Analysis, G.d.F., A.G., A.S., G.S.; Data Curation, G.L., G.P, N.V.; Writing—Original Draft Preparation, G.d.F., F.F.O., O.G.; Writing—Review & Editing, G.d.F., A.G., O.G., E.M., F.F.O., N.V, G.P., S.A.

**Funding:** This research received no external funding.

**Acknowledgments:** We are grateful to Les Greenberg for providing the transcripts of these cases.

**Conflicts of Interest:** The authors declare no conflict of interest.

## Appendix A

(1)   Canonical correlation coefficients between patient and therapist descriptors. Canonical Correlation is a way of inferring information from cross-covariance matrices. In the case we have two vectors $X = (X1, \ldots, Xn)$ and $Y = (Y1, \ldots, Ym)$ of random variables, and there are correlations among the variables, then canonical-correlation analysis finds linear combinations of the Xi and Yj which have maximum correlation with each other. Briefly, it measures the maximum interrelation between patient and therapist in a given time point. It is a "correlational-spectrum" analysis.

(2)   Percentage of explained variance by the first principal component. Very broadly used measure of order in a given system.

(3)   Sum of Pearson correlation coefficients higher than |0.25|. It turned out to be very effective in measuring the system's robustness and in predicting change and crises in economics [2].

(4)   Standard deviation of Pearson coefficients. The 2nd and 3rd are measures of order: the higher the measures (i.e., "% of Explained Variance" and "Gorban's G"), the more robust and connected the system's network. This (i.e., "Standard Deviation"), on the other hand, is a measure of dispersion. The higher the standard deviation, the more variable and flexible the system's network. The literature identifies extreme rigid or flexible network as dysfunctional systems.

(5)   Shannon Entropy on Eigenvalues. A commonly used measure of system order/disorder. A negative peak indicates a peak of system order and *vice versa*. It is a measure of "flatness" of the scree plot once a Principal Component Analysis is performed. A negative peak, on the other hand, indicates a steep slope on the scree plot [3].

From the application of such measures over the eight single cases, we have been able to show the dynamics of their psychotherapeutic processes over time (Figure A1).

From the inspection of Figure A1 it is clear that the quantitative routine allows the turning points of the series to often be easily detectable. In good-outcome cases (left), session 7 seems to be a crucial point in which a system change occurs, while in poor-outcome cases (right) this is true around session 9 or

14. It is also important to mention the difference between changes in component 1 and in component 2. A system's change in component 2 is described by the Shannon entropy and the variance explained by the first component (in orange and light-blue, respectively). Examples of these are session 7 for Margherita and session 14 for Secondolo. A system's change in component 1 is described by all the other measures and, for example, can be observed in session 7 for Sara, in session 7 for Lisa, and in session 9 for Primo.

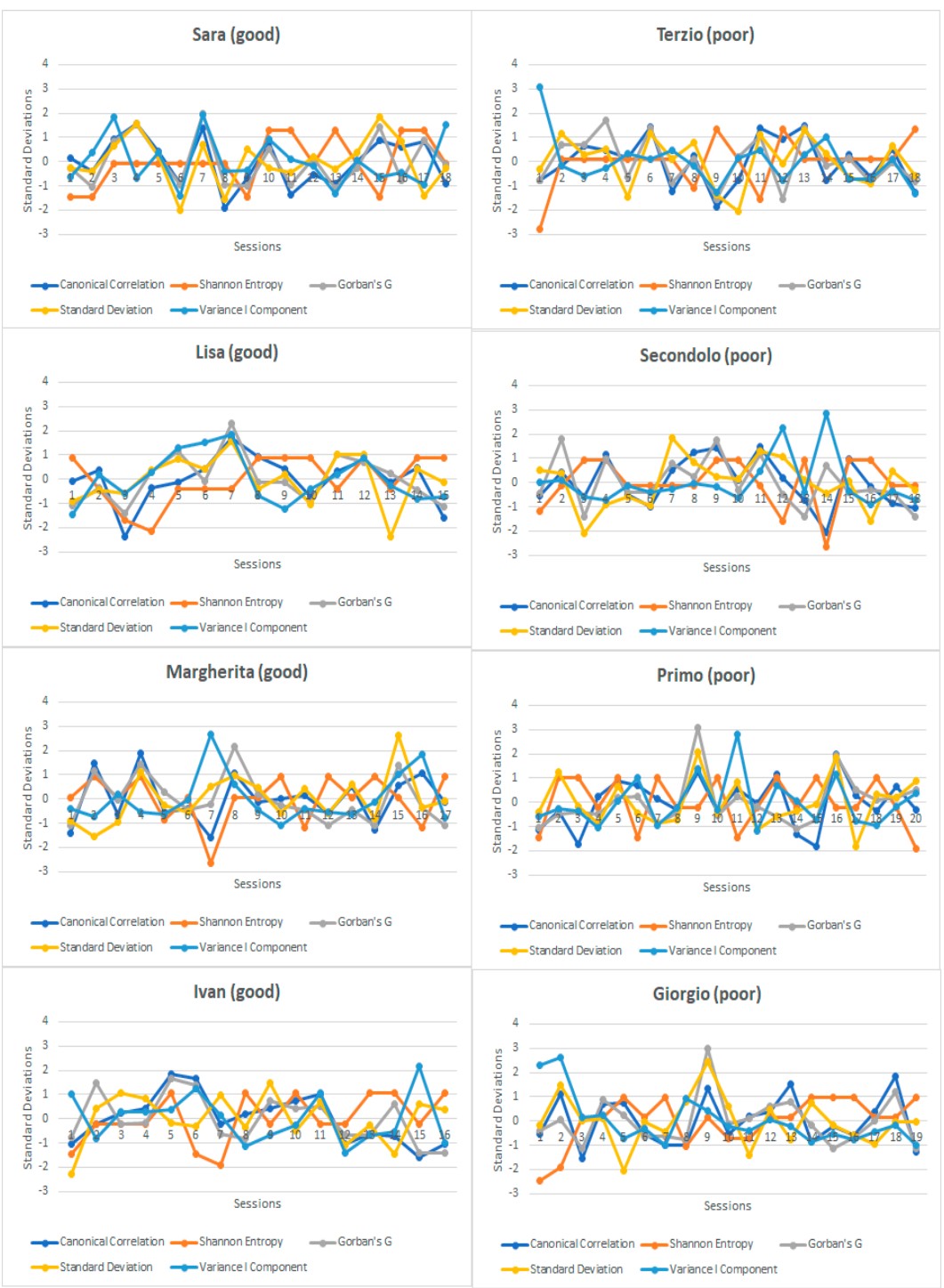

**Figure A1.** Dynamic analyses on the eight single cases (we use fictitious names to protect patients' confidentiality).

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
