# Peer review of "What Differentiates Poor and Good Outcome Psychotherapy? A Statistical-Mechanics-Inspired Approach to Psychotherapy Research"

_systems, doi:10.3390/systems7020022_

Round 1
Reviewer 1 Report
There is much to admire in this contribution. I have little to write concerning the manuscript itself. I do have some observations about future efforts. They are not essential to the review of this paper, however. The authors are to be commended for bringing a dynamical analysis to patient-therapist communication, and in particular for making connections to the work of Scheffer and others on dynamical measures that anticipate transitions.
For future consideration: Following Gorban (cited in the manuscript) one might suppose that increases in correlation and variance would precede clinically meaningful transitions. If the clinical state of the patient was measured longitudinally (perhaps administering the BDI after each session), it would be possible to address the question that is critical to the investigators thesis: do changes in the dynamics of patient-therapist communication precede clinical change?
As the authors know, disturbances of sleep are routinely observed in psychiatric populations. Longitudinal assessment of sleep quality (possibly the Pittsburgh Sleep Quality Index) might show that changes in the communications dynamic precede improvement in this critical psychiatric measure.
Similarly, a physiological measure could be considered. Measures of heart rate variabiltiy are typically reduced in depressed patients, and these measures can be obtained with minimal patient burden by measuring the ECG. Again the operation question becomes: do changes in the pattern of patient-therapist dialog precede changes in heart rate variability?
Explicit definitions (include equations) of measures listed on Page 4 would be helpful. In particular, what is the Shannon eigenvalue entropy? Is it the same as Morgera covariance complexity (Morgera, S. D. 1985. IEEE Transactions on Man, Systems and Cybernetics. 15, 608-619)?
Page 4 mentions an appendix. The version of the paper that I received for review did not have an appendix.
Reference 23 does not appear in the bibliography.
Author Response
There is much to admire in this contribution. I have little to write concerning the manuscript itself. I do have some observations about future efforts. They are not essential to the review of this paper, however. The authors are to be commended for bringing a dynamical analysis to patient-therapist communication, and in particular for making connections to the work of Scheffer and others on dynamical measures that anticipate transitions.
For future consideration: Following Gorban (cited in the manuscript) one might suppose that increases in correlation and variance would precede clinically meaningful transitions. If the clinical state of the patient was measured longitudinally (perhaps administering the BDI after each session), it would be possible to address the question that is critical to the investigators thesis: do changes in the dynamics of patient-therapist communication precede clinical change?
We thank the reviewer for his/her comment. This is exactly our research focus and we do want to proceed along this research line. The work empirically shows the possibility to reduce the complexity of the therapeutic dyad into two main dimensions (PC1 and PC2) of order-variability and elementary-complex. In a next paper we plan to try to build networks of successful and unsuccessful cases in order to check for differences in wiring architecture.
The inclusion of a clinical measure to establish its link with variations of correlation and variance is a really fruitful suggestion. This approach could be applied after a symbolic dynamics transformation of the data. This could help to study the temporal evolution of linguistic configurations of both patient and therapist. In this way we could be able to establish a link between the emergence of a new relational state (joint linguistic configuration of patient and therapist) and preceding variations of correlation and variance.
As the authors know, disturbances of sleep are routinely observed in psychiatric populations. Longitudinal assessment of sleep quality (possibly the Pittsburgh Sleep Quality Index) might show that changes in the communications dynamic precede improvement in this critical psychiatric measure.
Similarly, a physiological measure could be considered. Measures of heart rate variabiltiy are typically reduced in depressed patients, and these measures can be obtained with minimal patient burden by measuring the ECG. Again the operation question becomes: do changes in the pattern of patient-therapist dialog precede changes in heart rate variability?
Thank you for the very helpful suggestion.
Explicit definitions (include equations) of measures listed on Page 4 would be helpful. In particular, what is the Shannon eigenvalue entropy? Is it the same as Morgera covariance complexity (Morgera, S. D. 1985. IEEE Transactions on Man, Systems and Cybernetics. 15, 608-619)?
The Shannon eigenvalue entropy derives from the paper by (Sabatini, A. M. (2000). Analysis of postural sway using entropy measures of signal complexity. Medical and Biological Engineering and Computing, 38(6), 617-624.) and corresponds to the formula of Shannon having as p(i) the proportion of variance explained by the principal components.
We quoted the reference in the new version, this index is practically coincident with Morgera but is is applied on the eigenvectors of the correlation and not covariance matrix.
In the Appendix we included the measures definitions. (Now the Appendix that was absent in the previous version was restored).
Page 4 mentions an appendix. The version of the paper that I received for review did not have an appendix.
We are sorry, the Appendix was lost for a clerical error, now is included in the new verion of the manuscript.
Reference 23 does not appear in the bibliography.
We fixed the error.
Reviewer 2 Report
Dear authors
despite your article deals with a very crucial issue and offer an iintersting point of view,a few concerns must be taken into account:
1) Introduction: I would appreciate a more extensive review of the previous attempt to explore psychotherapy effectiveness, both in terms of processes and of systems of meaning.
2) Methodology: I have two main concerns. On the one hand, there is not a clear-cut analysis of the limits and constraints of your methodology, by especially considering that many attempts to evaluate effectiveness exist and the Mergenthaler’s Therapeutic Cycle Model cannot be considered a standard procedure, or better we do not have many studies to clearly evaluate pros and cons of such an approach. On the other hand, when considering an attempt to develop a "systemic" view at statistical analysis of effectiveness, PCA may turn out to be biased (e.g. sampling error). Modern psychopathology frequently offers network analyses approach I woul consider (either to improve your methods or as cue for discussion).
3) Discussion: I would suggest to better explore limits and weaknesses of your study. On the one hand, the specific types of methods, teherapy and sample size may affect the generalizability. On the other hand, such variables may have significantly channalized the results you may find yourselves analyzing these biases rather than prove that TCM components are usuful in evaluating effectiveness and outcomes.
I hope these suggestions may help you improving your paper.
Author Response
Dear authors
despite your article deals with a very crucial issue and offer an interesting point of view, a few concerns must be taken into account:
1) Introduction: I would appreciate a more extensive review of the previous attempt to explore psychotherapy effectiveness, both in terms of processes and of systems of meaning.
We thank the reviewer for his/her appreciation of our issue. We followed the suggestion to enlarge the introduction and modified the text accordingly, including new references.
2) Methodology: I have two main concerns. On the one hand, there is not a clear-cut analysis of the limits and constraints of your methodology, by especially considering that many attempts to evaluate effectiveness exist and the Mergenthaler’s Therapeutic Cycle Model cannot be considered a standard procedure, or better we do not have many studies to clearly evaluate pros and cons of such an approach.
Thank you for this comment, we would like to highlight the below reference summarizing the strenghts and limitations of the applied coding system:
Mergenthaler, E. (2015). Resonating Minds: A School-Independent Theoretical Conception and its Empirical Application to Psychotherapeutic Processes. In B. M. Strauss, J. P. Barber, & L. G. Castonguay (Eds.), Visions in Psychotherapy Research and Practice. Reflections from Presidents of the Society for Psychotherapy Research (pp. 292-314). New York and London: Rutledge.
In any case, the adopted coding system, broadly used in previous years in psychotherapy research, has been adopted only for the transformation of linguistic variables into quantitative elements. The main innovative aspect lay in the investigation of those quantitative elements of language through few coarse grained dimensions of order-variability (PC1) and elementary-complex (PC2).
On the other hand, when considering an attempt to develop a "systemic" view at statistical analysis of effectiveness, PCA may turn out to be biased (e.g. sampling error). Modern psychopathology frequently offers network analyses approach I would consider (either to improve your methods or as cue for discussion).
We thank the reviewer for his/her suggestion. This is the first part of a larger project, including a second part in which we will apply network tools. In this work, PCA is instrumental to higlight the shift of meaning of the same codes in good and poor outcome cases in terms of different correlation structures. This is the most straightforward method to detect such a change in correlation/meaning and it is widespread in many different science fields (Wei, X., & Li, K. C. (2010). Exploring the within-and between-class correlation distributions for tumor classification. Proceedings of the National Academy of Sciences, 107(15), 6737-6742.).
In this work, we focused on the possibility to reduce the complexity of the therapeutic dyads into few general dimensions of order-variability (PC1) and elementary-complex (PC2). We demonstrated the statistical significance of these dimensions potentially applicable to different dataset grounded both on language and physiology (e.g. ECG, EEG, galvanic skin response).
We included in the conclusion a short paragraph stating that this manuscript is preparatory to the second based on network analyses.
3) Discussion: I would suggest to better explore limits and weaknesses of your study. On the one hand, the specific types of methods, teherapy and sample size may affect the generalizability. On the other hand, such variables may have significantly channalized the results you may find yourselves analyzing these biases rather than prove that TCM components are usuful in evaluating effectiveness and outcomes.
We changed the text accordingly (see Conclusion).
I hope these suggestions may help you improving your paper.
We thank you for the precious suggestions.
Reviewer 3 Report
Tipping points in social systems (including psychotherapy dyads) have been widely investigated using dynamical systems theory and complexity science, and more references of this would be appropriate. Likewise, the concept is introduced but no tipping points / critical points / bifurcations are highlighted in the analysis or conclusion. Would like to see that.
Complete speaking turns don't usually break up into exactly 150-word blocks, which could cause the analytic tool employed to misconstrue affect of a client vs. therapist. How do we know the analysis is not a spurious product of arbitrary transcript reduction? What happens when one side of an exchange is longer than 150 words? Or shorter? How could it be done differently?
Author Response
Tipping points in social systems (including psychotherapy dyads) have been widely investigated using dynamical systems theory and complexity science, and more references of this would be appropriate.
We thank the reviewer for the comment, we modified the text accordingly adding a short review of previous complexity-based approaches in the Introduction.
Likewise, the concept is introduced but no tipping points / critical points / bifurcations are highlighted in the analysis or conclusion. Would like to see that.
The reviewer is totally right, as a matter of fact we did not find any evidence of such transitions in our data set. This derives from the brevity of those therapeutic processes (i.e. brief psychotherapies with a mean of 17 sessions). This kind of dataset is not sufficient to observe such events.
Complete speaking turns don't usually break up into exactly 150-word blocks, which could cause the analytic tool employed to misconstrue affect of a client vs. therapist. How do we know the analysis is not a spurious product of arbitrary transcript reduction? What happens when one side of an exchange is longer than 150 words? Or shorter? How could it be done differently?
In the software TCM there is an option called “turn” that moves the word block boundary to the nearest turn of speech (+ or - 15 words). This allowed us to eliminate the 'border effects' the reviewer refers to.
Round 2
Reviewer 2 Report
Dear editor
the paper has not been significantly improved after the revision. Thus, I confirm my suggestion to reject the paper.
Regards